# Exploring the Link between Family Health and Health Literacy among College Students: The Mediating Role of Psychological Resilience

**DOI:** 10.3390/healthcare11131859

**Published:** 2023-06-26

**Authors:** Yan-Yan Wang, Xin-Cheng Huang, Jie Yuan, Yi-Bo Wu

**Affiliations:** 1School of Economics and Management, Beijing Institute of Graphic Communication, Beijing 102600, China; w2544798727@163.com (Y.-Y.W.); 18883779527@163.com (X.-C.H.); 2Jitang College of North China University of Science and Technology, Tangshan 063210, China; 3School of Public Health, Peking University, Beijing 100080, China

**Keywords:** family health, health literacy, psychological resilience, undergraduates

## Abstract

Enhancing health literacy is of the utmost importance to enhance the physical and mental well-being of college students. Unfortunately, there has been limited research investigating the means of improving college students’ health literacy through the perspective of families. Family health is an interdisciplinary and complex concept that involves multiple factors, and it provides a holistic perspective on the overall well-being of the family unit. Thus, this study aims to examine the relationship between family health and health literacy and scrutinize the mediating role of psychological resilience. A valid sample of 5473 students was collected from a university in November–December 2022 and was assessed using regression analysis. The findings reveal that family health has a positive association with the health literacy of college students (*β* = 0.56, *p* < 0.001), with psychological resilience playing a critical mediating role (*β* = 0.11, 95% CI: [0.09, 0.13]). Therefore, the family ought to be recognized as a fundamental mechanism to enhance college students’ health literacy. Additionally, it is essential to emphasize the amelioration of psychological distress among college students and enhance their psychological resilience, which will be helpful for their overall health consciousness and proficiency.

## 1. Introduction

Health literacy plays a pivotal role in individuals’ physical and psychological well-being [1]. Insufficient health literacy contributes to inferior levels of individual physical and mental health and gives rise to a range of profound societal issues, such as health disparities [2]. Cultivating robust health literacy necessitates a protracted and concerted effort, particularly during the formative years. It is imperative that individuals, particularly in their youth, actively cultivate wholesome habits and augment their capacity for health. College students are at a prime stage of learning and it is crucial for them to gain comprehensive health competence, develop health consciousness, and adopt healthy behaviors [3]. Furthermore, the health status of college students not only impacts their own future physical and psychological well-being but also holds significant ramifications for societal and national development [4].

According to the 2020 Chinese College Student Health Survey, overall health conditions among Chinese college students are generally favorable. However, a noteworthy segment of college students still struggles with oral diseases, gastrointestinal ailments, dermatological disorders, and diverse psychological disturbances resulting from unhealthy lifestyles and inadequate health awareness [5]. Insufficient health literacy is a key factor contributing to unfavorable health behaviors and suboptimal health outcomes among college students [6]. Health literacy is important in determining psychological disorders and quality of life among college students. Improving health literacy levels can help reduce the risk of common psychological issues, including perceived stress, depressive symptoms, and impulsivity, while concurrently improving quality of life [7]. Conversely, limited health literacy may exert an impact on critical behavioral and health outcomes [8], encompassing obesity [9] and smoking habits [10]. Hence, enhancing college students’ health literacy assumes paramount importance in relation to their physical well-being, as well as for the betterment of society and the nation at large.

Health literacy is commonly defined as the extent to which an individual possesses the capacity to access, comprehend, and utilize fundamental health information and services, enabling them to make informed health-related decisions [11]. Enhancing health literacy has become a significant topic of research [2,12,13]. Existing studies show that factors such as gender [14], educational attainment [15], income level [16], and physical and psychological well-being [14] have a significant impact on individuals’ health literacy levels. Furthermore, apart from examining individual characteristics, researchers have explored the factors impacting health literacy through the application of theoretical frameworks, including the health belief model, social cognitive theory, and stage change theory [17]. Regarding strategies for improving health literacy, experts widely agree that improving information dissemination, fostering effective communication, and implementing structured educational interventions constitute effective approaches to enhancing individual health literacy [18].

Nonetheless, the influence of family on health literacy is a significant factor that has garnered relatively less attention. Family members play a crucial role in promoting a healthy lifestyle by facilitating access to health resources, fostering health-related skills, and enhancing comprehension of health-related matters [19]. Grounded in family systems theory, the family is conceptualized as an interconnected system wherein members exchange information, perspectives, and emotions, and thereby influence one another [20]. Although college students begin transitioning away from the family environment and embark on a journey towards independence and social adaptation, their connection with their families remains significant. With the widespread use of mobile devices, students frequently communicate with their family members [21]. Research indicates that individuals benefiting from higher levels of family involvement in health matters are more likely to receive support and care related to their health from their familial connections. Such support contributes to the enhancement of their health literacy levels, consequently reducing the likelihood of developing mental health issues [22]. When families are larger and possess greater resources, individuals tend to acquire health-related knowledge from their family members. Moreover, the younger generations exhibit a heightened motivation to acquire such knowledge [23].

Family health can be conceptualized as a collective asset stemming from the well-being of each family member, their interactions and capabilities, and the convergence of the family’s physical, social, emotional, economic, and medical resources [24]. Health literacy extends through familial and social networks as a shared resource, with individuals often relying on the health literacy skills of their family members to seek, comprehend, and utilize health-related information. This dynamic allows for compensation when there are limitations in individual health literacy since other family members with higher levels of health literacy can provide support. Consequently, the health literacy of college students who are embedded within a familial context exhibits a profound interdependence with the overall health status of the family [25]. The health of the family, in turn, profoundly impacts the capacity of college students to access, comprehend, and effectively employ health information and services in order to enhance their own well-being [26].

Psychological resilience refers to the adaptive process through which an individual responds to threats, tragedies, adversity, trauma, or significant stress, thereby demonstrating the ability to recover and sustain psychological well-being in the face of difficulties [27]. As a dynamic framework for withstanding adverse environmental influences, psychological resilience emerges through the interplay of protective factors from the individual, family, and society [28]. Consequently, the family serves as a potential safeguarding system in relation to psychological resilience. Research indicates that a supportive, nurturing, and stable family environment promotes individual resilience while reducing negative emotions such as depression and anxiety [29]. Moreover, close relationships and support networks among family members facilitate effective coping with challenges and stress, thereby enhancing psychological resilience [30].

Psychological resilience is closely connected to both health competence and health status. Research has demonstrated that individuals with higher levels of psychological resilience exhibit greater mental health literacy, thereby enhancing their overall mental well-being [31]. According to family systems theory, the family, functioning as an emotional system, has the power to influence the psychological state of its members, subsequently influencing the health outcomes of individuals [1]. As a supportive system, the family has the potential to enhance and foster the levels of psychological resilience among college students, thereby strengthening their ability to acquire health-related knowledge and skills.

The literature concerning the association between family health, psychological resilience, and health literacy among college students is extremely scarce. Therefore, the purpose of this study was to examine the relationship between family health and college students’ health literacy and the mediating role that psychological resilience plays in this relationship. Based on the previous analysis, we proffer two hypotheses to investigate the factors that are linked to health literacy in college students, while the hypothesis model is presented in Figure 1.

**Hypothesis 1 (H1).** *Family health has a positive relationship with health literacy among college students*.

**Hypothesis 2 (H2).** *Psychological resilience operates as a mediator in the relationship between family health and health literacy among college students*.

## 2. Materials and Methods

### 2.1. Procedures and Participants

We carried out a cross-sectional investigation among current undergraduate students at the Jitang College of North China University of Science and Technology in Hebei, China, from 12 November 2022 to 13 December 2022; this survey was conducted among all undergraduate students in the university. The inclusion criteria for this study were: (1) currently enrolled in this college as an undergraduate student, (2) voluntary participation in this research, and (3) non-participation in other similar studies. The exclusion criteria were: (1) incomplete completion of the questionnaire, and (2) a questionnaire completion time of less than 60 s (we considered the quality of questionnaires completed in less than 60 s to be unreliable). To ensure quality control and minimize bias, we attempted to include all the students in the university to mitigate the impact of non-representativeness. Additionally, the investigators received training and passed an assessment before participating in the study, and regular assessments and sampling of the investigators’ work were conducted. In the data analysis phase, we employed stratified analysis and other methods to reduce the confounding bias. With the help of the college’s teachers, we aimed to collect 5495 samples; eventually, we successfully collected 5475 samples (reasons for non-participation included an unwillingness to participate in the study and hospitalization due to illness). The effective recovery rate was 96.9%. After excluding 2 samples with missing data, 5473 samples were included in our analysis. This study has been approved by the Ethics Review Committee of the Jitang College of North China University of Science and Technology, under the approval number: JTXY-2022-002.

### 2.2. Measures

Demographic information: In this study, we included demographic variables such as age, gender, place of residence, monthly per capita household income, and each parent’s highest education level.

Family health: The Family Health Scale (FHC) was developed by Crandall et al. [24]. We used the short version of this scale (FHC-SF), which contains four dimensions: the family’s social and emotional health processes, the family’s healthy lifestyle, the family’s socioeconomic resources, and the family’s emotional resources. The Family Health Scale–Short Form is useful as a uniform measure of family health and has good reliability and validity in Chinese populations [32]. The scale we used consists of 10 items, such as, “In my family, we support each other”, “In my family, we maintain hope even in very difficult times”, and “In the past 12 months, my family’s housing has not been able to meet the needs of the family”, while items 6, 9, and 10 were reverse-encoded. Each item was rated on a 5-point Likert scale ranging from 1 to 5 (strongly disagree to strongly agree) (Cronbach’s α = 0.827, M = 37.98, SD = 6.81), yielding a cumulative score of between 10 and 50, wherein higher scores indicate a greater degree of family health.

Psychological resilience: Psychological resilience refers to positive adaptation in the face of stress or trauma. The Connor–Davidson Resilience Scale (CD-RISC) is used to measure resilience and is a widely used instrument that was developed by Connor and Davidson [33]. The original version of this scale has 25 items, each of which is rated on a 5-point scale ranging from 0 to 4 (“not true at all” to “true nearly all the time“). We used the 10-item version of the Connor–Davidson Resilience Scale, which was previously validated in the Chinese context [34]. The Chinese version of the 10-item CD-RISC has excellent psychometric properties and is applicable to Chinese people. The total score ranges from 0 to 40 (Cronbach’s α = 0.941, M = 32.54, SD = 8.06), with a higher score indicating a higher level of psychological resilience.

Health Literacy: The Health Literacy Scale–Short Form (HLS-SF) was devised by Duong et al. [35]. In our study, we employed the simplified version of the HLS-SF, which has demonstrated strong reliability and validity among the Chinese population [36]. The simplified version consists of three domains—health care, disease prevention, and health promotion—with each domain comprising three inquiries, resulting in a total of nine items. Participants were requested to self-report the ease with which they could undertake various health-related tasks, such as “procuring information concerning the management of your ailment” and “comprehending the instructions provided with your medication”. Responses were rated on a 4-point Likert scale ranging from 0 to 3 (very difficult to very easy) (Cronbach’s α = 0.972, M = 24.28, SD = 6.86), yielding a cumulative score of between 0 and 27, wherein higher scores indicate a greater degree of health literacy.

### 2.3. Statistical Analysis

In this study, we utilized SPSS version 25.0 and Process version 3.0 to conduct data analysis and testing, assuming a two-tailed significance level of 0.05. Firstly, descriptive statistics (mean, standard deviation, and sample size/proportion) and correlation tests were conducted using SPSS, with the results of the descriptive statistics being presented in Table 1, and the results of the variable correlation tests being presented in Table 2. Secondly, multiple regression analysis was performed to test the relationships between the independent variable, Family Health, the mediating variable, Psychological Resilience, and the dependent variable, Health Literacy. Specifically, H1 was tested, and the results of the regression analysis can be found in Table 3. Finally, to test for mediation effects, we used the Process method developed by Hayes, setting the Bootstrap number to 5000 with a 95% confidence interval significant level to verify H2. The results of the mediation effect test are presented in Table 4. Additionally, demographic variables such as age, gender, place of residence, monthly per capita household income, and highest parental education level were included in the regression equation as control variables.

## 3. Results

### 3.1. Results of Descriptive Statistics

The mean standard deviation of age among all samples was 20.53 ± 1.60 years. The proportion of male individuals was 40.0%, totaling 2189 individuals. The proportion of individuals residing in rural areas was 46.6%, totaling 2551 individuals. The proportion of households with a per capita monthly income of below RMB 6000 was 77.2%, totaling 4225 households. The percentage of fathers with their highest education level being at high school or below was 79.8%, totaling 4368 individuals. The percentage of mothers with their highest education level being at high school or below was 81.7%, totaling 4473 individuals.

### 3.2. Correlation Test between Variables

Based on the information provided in Table 2, the mean and standard deviation for family health was 37.98 ± 6.81, that for psychological resilience was 32.54 ± 8.06, and that for health literacy was 24.28 ± 6.86. Notably, the results indicate moderate and significant correlations between family health and psychological resilience (*r* = 0.396, *p* < 0.001) and between family health and health literacy (*r* = 0.447, *p* < 0.001), as well as between psychological resilience and health literacy (*r* = 0.558, *p* < 0.001).

### 3.3. Regression Analysis Results and Mediating Effect Test

The results of the regression analyses demonstrated a positive association between family health and health literacy (*β* = 0.56, *p* < 0.001), even after adjusting for variables such as age, gender, place of residence, monthly per capita household income, and the highest level of parental education. As such, the findings support H1. Additionally, the analyses revealed that family health had a positive impact on psychological resilience (*β* = 0.40, *p* < 0.001), which, in turn, positively influenced health literacy (*β* = 0.27, *p* < 0.001). Therefore, H2 was also supported. Furthermore, the combined explanatory capacity of demographic, independent, and mediating variables on health literacy amounted to 37%. 

The findings of the mediating effect analysis are displayed in Table 4, revealing that the significant indirect impact of family health on health literacy is channeled through psychological resilience (*β* = 0.11, 95% CI: [0.09, 0.13]), with an indirect effect percentage of 18.93%.

## 4. Discussion

Health literacy has a significant impact on both physical and mental health, highlighting the importance of improving health literacy for individuals. This research aimed to explore the relationship between family health, psychological resilience, and health literacy in college students from a family perspective. The study’s findings revealed that the mean proficiency of health literacy, psychological resilience, and health literacy among Chinese university students exhibited a notably elevated level. This suggests that the present health literacy standard among college students has reached a significant threshold, which is consistent with the observation that the group of individuals aged 25–34 years has the highest level of health literacy, as reported in the 2020 national health literacy monitoring report [37]. Meanwhile, the findings also suggest that family health is significantly associated with the health literacy of college students, while psychological resilience may act as a mediator between family health and health literacy.

The outcomes of our study indicate that family health has a positive and significant relationship with health literacy. This implies that college students with higher levels of family health may possess greater levels of health literacy. According to the circular model of marriage and family systems theory proposed by Olson et al., when families function more optimally, they are better equipped to foster positive attitudes and behaviors in their members [38]. Consequently, with healthier family functioning, family members can better empathize with and assist one another, leading to increased awareness of health and healthier behaviors. This finding aligns with previous research findings. For instance, a study conducted in the Philippines explored the influence of family environment on adolescent health behaviors and found that a healthy family environment significantly enhanced adolescents’ health behaviors by means of open and positive communication and mutual respect among family members [39]. Families serve as critical protective factors for college students’ health problems. Support from their families, cohesive family relationships, and positive family communication are essential factors for enhancing health awareness and health literacy in college students. Families provide students with more social support and resources and help them to alleviate the health and psychological stressors that they encounter during adolescence [40]. Nonetheless, it is important to acknowledge that the health literacy assessed in this study is based on self-reporting, which may not accurately reflect the actual health literacy proficiency of university students. Nevertheless, we maintain the belief that advocating for an increased emphasis on familial intervention tools to address the health concerns of college students is justified. The aim of such advocacy is to promote the development of higher health literacy levels among this demographic, through measures such as strengthening familial relationships, enhancing family resources, and improving familial lifestyles.

The study revealed that psychological resilience mediates the relationship between family health and health literacy. Specifically, individuals with healthier families tend to demonstrate higher levels of psychological resilience, enabling them to cope more effectively with various stressful life events, including health-related challenges. Thus, psychological resilience plays a crucial role in enhancing individuals’ health literacy levels. Previous research supports the present study’s findings. For instance, Chew et al. found that family support was a crucial factor in increasing psychological resilience among young people because strong family bonds could offer appropriate support when required, thereby enhancing resilience when coping with the stresses and challenges associated with epilepsy [41,42,43]. Moreover, family communication contributes to the restoration of psychological resilience in post-traumatic adolescents, enabling them to cope better with health-related life events [44]. Chen and Harris also found that positive family relationships help cultivate a state of psychological well-being in adolescents, which can ultimately improve their health [45]. In summary, this study suggests that family health can promote a positive mental health status among college students, leading to enhanced resilience when coping with stressful or traumatic events, ultimately improving their health literacy skills. Therefore, we emphasize the importance of constructing a family-centered physical and psychological resilience treatment program for college students. Such a program is essential for both enhancing their health competencies and mitigating health problems. 

## 5. Implications and Limitations

The present study carries both theoretical and practical implications. Firstly, the study highlights the significance of family health as an important factor in improving individual health literacy. The findings emphasize the importance of family interventions as a means of improving the health behaviors and perceptions of college students. Policymakers and medical professionals should consider the family health status of college students and implement family-centered interventions to improve their health literacy and overall health status. Secondly, the study underscores the mediating role of psychological resilience in the relationship between family health and health literacy. Therefore, researchers and medical professionals should focus on the mental health status of family members, especially psychological resilience in the face of adversity. They should employ psychotherapy and other means by which to develop and enhance the psychological resilience of college students, while considering family interventions to boost their self-confidence and pay attention to their physical and mental health. This approach will contribute to improving the overall health literacy level of college students.

This study is not without its limitations, as there are several points that need to be addressed in future research. Firstly, the study adopted a single-center cross-sectional survey, which cannot establish a causal relationship between family health, psychological resilience, and health literacy. To overcome this limitation, future studies may consider using nationally representative large-sample surveys or longitudinal studies. Secondly, the Family Health Short Form used in the study can only assess the overall level of family health and cannot explore the impact of various aspects of family health on individual health literacy. Future studies may use more comprehensive measures to investigate the specific factors that contribute to the relationship between family health and health literacy. 

## 6. Conclusions

This research investigated the correlation between familial health, psychological resilience, and the health literacy of college students. The findings suggest that family health may be a potential key factor in improving college students’ health literacy, whereas psychological resilience functions as a mediating mechanism. Ultimately, the family unit ought to be regarded as a crucial instrument to boost college students’ health literacy. Additionally, greater emphasis should be placed on addressing the psychological distress of college students and enhancing their psychological resilience, which will inevitably augment their health consciousness and health proficiency.

## Figures and Tables

**Figure 1 healthcare-11-01859-f001:**
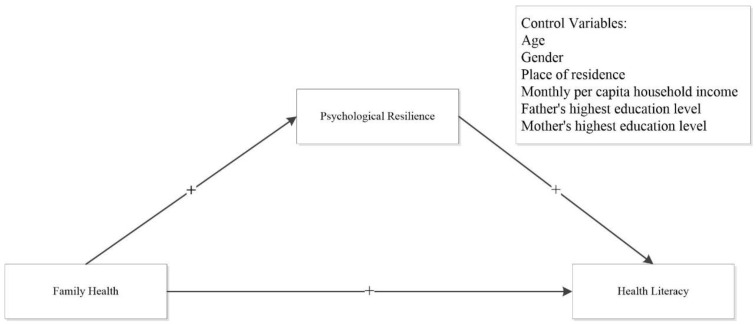
Hypothesis model.

**Table 1 healthcare-11-01859-t001:** Descriptive statistics of the participants (*N* = 5473).

Variables		M ± SD or *n* (%)
Age		20.53 ± 1.60
Gender		
	Male	2189 (40.0)
	Female	3284 (60.0)
Place of residence		
	Rural area	2551 (46.6)
	Town	2922 (53.4)
Monthly per capita household income (CNY)		
	≤1000	699 (12.8)
	1001–2000	807 (14.7)
	2001–3000	835 (15.3)
	3001–4000	728 (13.3)
	4001–5000	630 (11.5)
	5001–6000	526 (9.6)
	6001–9000	480 (8.8)
	9001–12,000	341 (6.2)
	12,001–15,000	159 (2.9)
	≥15,000	268 (4.9)
Father’s highest education level		
	No formal education	181 (3.3)
	Elementary school	900 (16.4)
	Junior high school	1975 (36.1)
	Junior college	497 (9.1)
	High school	815 (14.9)
	College	539 (9.8)
	Undergraduate	483 (8.8)
	Master	40 (0.7)
	PhD	43 (0.8)
Mother’s highest education level		
	No formal education	274 (5.0)
	Elementary school	1119 (20.4)
	Junior high school	1914 (35.0)
	Junior college	520 (9.5)
	High school	646 (11.8)
	College	503 (9.2)
	Undergraduate	427 (7.8)
	Master	25 (0.5)
	PhD	45 (0.8)

Note. The exchange rate of USD to CNY as of 20 June 2023 is: 1 USD = 7.18 CNY.

**Table 2 healthcare-11-01859-t002:** Descriptive statistics and correlation analysis of all variables.

	M (SD)	(1)	(2)	(3)	(4)	(5)	(6)	(7)	(8)	(9)
(1)	20.53 (1.60)	1								
(2)	0.60 (0.49)	−0.040 **	1							
(3)	0.53 (0.50)	−0.078 ***	−0.056 ***	1						
(4)	4.43 (2.55)	−0.127 ***	−0.097 ***	0.303 ***	1					
(5)	3.89 (1.72)	−0.073 ***	−0.077 ***	0.380 ***	0.344 ***	1				
(6)	3.69 (1.73)	−0.102 ***	−0.060 ***	0.413 ***	0.331 ***	0.671 ***	1			
(7)	37.98 (6.81)	−0.160 ***	0.169 ***	0.064 ***	0.152 ***	0.094 ***	0.096 ***	1		
(8)	32.54 (8.05)	−0.037 ***	0.036 ***	0.049 ***	0.080 ***	0.076 ***	0.090 ***	0.395 ***	1	
(9)	24.28 (6.85)	−0.064 ***	0.085 ***	0.063 ***	0.061 ***	0.071 ***	0.078 ***	0.557 ***	0.446 ***	1

Note. (1) Age; (2) gender; (3) place of residence; (4) monthly per capita household income; (5) father’s highest education level; (6) mother’s highest education level; (7) family health (range: 5–50); (8) psychological resilience (range: 0–40); (9) health literacy (range: 0–27); ** *p* < 0.01; *** *p* < 0.001.

**Table 3 healthcare-11-01859-t003:** Model of the regressions among family health, psychological resilience, and health literacy.

Regression (*N* = 5473)		Fitting Metrics			Coefficients and Significances	
Outcome variables	Predicting variables	*R*	*R* ^2^	*F*(*df*)	*β*	*t*
Health literacy		0.56	0.31	498.07 *** (7)		
	Age				0.03	2.27 *
	Gender				−0.01	−0.92
	Place of residence				0.03	2.24 *
	Monthly per capita household income				−0.04	−3.29 **
	Father’s highest education level				0.005	0.35
	Mother’s highest education level				0.02	1.57
	Family health				0.56	48.39 ***
Psychological resilience		0.40	0.16	149.13 *** (7)		
	Age				0.03	2.51 *
	Gender				−0.03	−2.10 *
	Place of residence				0.002	0.12
	Monthly per capita household income				0.002	0.17
	Father’s highest education level				0.006	0.32
	Mother’s highest education level				0.05	2.78 **
	Family health				0.40	31.05 ***
Health literacy		0.61	0.37	406.63 *** (8)		
	Age				0.02	1.60
	Gender				−0.003	−0.31
	Place of residence				0.03	2.30 *
	Monthly per capita household income				−0.04	−3.50 ***
	Father’s highest education level				0.004	0.27
	Mother’s highest education level				0.01	0.79
	Family health				0.46	37.92 ***
	Psychological resilience				0.27	22.79 ***

Note. * *p* < 0.05; ** *p* < 0.01; *** *p* < 0.001.

**Table 4 healthcare-11-01859-t004:** Analysis of the total, direct, and mediating effects.

	Estimate	Bootstrap SE	Bootstrap 95% CI		Effect Ratio
			Lower	Upper	
Total Effect	0.56	0.01	0.55	0.59	-
Direct Effect	0.46	0.01	0.44	0.48	81.67%
Mediating Effect	0.11	0.01	0.09	0.13	18.93%

Note. Bootstrap SE and Bootstrap 95% CI refer to the standard error and the lower and upper limits of the 95% confidence interval of the indirect effects, estimated by the bias-corrected percentile Bootstrap method, respectively; all values are rounded to two decimal places.

## Data Availability

Data are available, upon reasonable request, by emailing: bjmuwuyibo@outlook.com.

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
