# Peer review of "Exploring the Link between Family Health and Health Literacy among College Students: The Mediating Role of Psychological Resilience"

_healthcare, 2023, doi:10.3390/healthcare11131859_

Round 1

Reviewer 1 Report

Building Health through Family Ties: Exploring the Link between Family Health and Health Literacy among College Students, Investigating the Mediating Role of Psychological Resilience

This is an interesting contribution to the journal, aiming at examining the relationship between family health and health literacy assessing the mediating role of psychological resilience with a large sample of participants. Still, I believe some minor changes would improve the overall quality of the article:

1.     Please provide clear objectives of the study before the hypotheses statement.

2.     Please provide further information on the whole-group sampling technique.

3.     Was the FHC used validated for China?

4.     Current version of SPSS is 29, why did you use v.25?

5.     We need to know overall measures of family health, health literacy and psychological resilience before further results.

6.     Please provide a figure depicting the mediation results.

Best wishes.

Reviewer 2 Report

The study presented in the manuscript seeks to explore the relationship between family health and family literacy among college students, particularly analyzing the mediating effect of psychological resilience.

The study is based on a very large sample size and is well-articulated. However, the text seems at times awkward to read, thus I would suggest that the authors pay more attention to English fluency.

Below are a few comments regarding the main text.

The title is a bit long and might be shortened while keeping its expressive efficacy.

In the Introduction, at lines 98-102, the authors here put forward a few statements without quoting any relevant piece of literature. Who supports these claims?

Lines 311-312: The authors state, “The findings reveal the crucial psychological mechanisms through which families can influence individual health.” I would rather speak of resilience only, since other “psychological mechanisms” were not examined in the study.

I detected a typo at line 168, where the authors should write a comma, not a period.

The English is at times difficult to read. I suggest that the authors perform a moderate English revision.

Reviewer 3 Report

This Reviewer offers the following recommended modifications to the manuscript:

Page 1, Abstract:  Abstract is well written and conveys to the Reader the relevance and importance of this topic.

 Page 3, Line 126:  The Authors state that “the hypothesis model is presented in figure 1:”  Recommend the Authors capitalize the word “figure” for accuracy. The new sentence should read as follows: “and the hypothesis model is presented in Figure 1:”

 Page 4, Line 142: The Authors state that one of the exclusion criterion is “questionnaire completion time less than 60 seconds.” This Reviewer is unclear as to why this is an exclusion criterion. Recommend the Authors elaborate in one sentence why this an exclusion criterion.

 Page 4, Line 171: The Authors state “The Connor–Davidson Resilience Scale (CD-RISC) is to measure  . . .”  Recommend the Authors insert the word “used” in this sentence for improved clarity to the Reader. The new sentence should read as follows: “The Connor–Davidson Resilience Scale (CD-RISC) is used to measure . . . .”

 Page 5, Line 198: The Authors state “regression analysis was performed to test the relationships between the independent variable, family Health  . . .”  Recommend the Authors capitalize both words “family and Health” for accuracy and consistency. The new sentence should read as follows: “regression analysis was performed to test the relationships between the independent variable, Family Health. . . .”

 Page 8, Line 273: The Authors state “. ..  health and psychological stresses. . .”  Recommend the Authors replace the word “stresses” with “stressors” for accuracy.  The new sentence should read as follows: “. ..  health and psychological stressors. . .” 

 Concluding comments: This Reviewer commends the Authors for a well-written analyses in the very important domain of health literacy, psychological resiliency and the importance of family for the college student population.  The study findings and recommendations shed light on an important area and provide great ‘best practices’ for academic institutions such as universities, to incorporate mental health services into their campus health clinics. This Reviewer looks forward to these minor revisions as well as to follow-on studies on this topic by these talented Authors.
